# Approaches for Memristive Structures Using Scratching Probe Nanolithography: Towards Neuromorphic Applications

**DOI:** 10.3390/nano13101583

**Published:** 2023-05-09

**Authors:** Roman V. Tominov, Zakhar E. Vakulov, Vadim I. Avilov, Ivan A. Shikhovtsov, Vadim I. Varganov, Victor B. Kazantsev, Lovi Raj Gupta, Chander Prakash, Vladimir A. Smirnov

**Affiliations:** 1Research Laboratory Neuroelectronics and Memristive Nanomaterials (NEUROMENA Lab), Institute of Nanotechnologies, Electronics and Electronic Equipment Engineering, Southern Federal University, Taganrog 347922, Russia; tominov@sfedu.ru (R.V.T.); zvakulov@sfedu.ru (Z.E.V.); avilovvi@sfedu.ru (V.I.A.); shihovcov@sfedu.ru (I.A.S.); varganov@sfedu.ru (V.I.V.); kazantsev@neuro.nnov.ru (V.B.K.); lovi.raj@lpu.co.in (L.R.G.); chander.mechengg@gmail.com (C.P.); 2Department of Radioelectronics and Nanoelectronics, Institute of Nanotechnologies, Electronics and Electronic Equipment Engineering, Southern Federal University, Taganrog 347922, Russia; 3Institute of Biology and Biomedicine, National Research Lobachevsky State University of Nizhny Novgorod, Nizhny Novgorod 603950, Russia; 4Division of Research and Development, Lovely Professional University, Phagwara 144411, Panjab, India; 5School of Mechanical Engineering, Lovely Professional University, Phagwara 144411, Panjab, India

**Keywords:** artificial intelligence, neuromorphic systems, memristor, ReRAM, resistive switching, scratching probe nanolithography, Taguchi method, ZnO thin films, pulsed laser deposition

## Abstract

This paper proposes two different approaches to studying resistive switching of oxide thin films using scratching probe nanolithography of atomic force microscopy (AFM). These approaches allow us to assess the effects of memristor size and top-contact thickness on resistive switching. For that purpose, we investigated scratching probe nanolithography regimes using the Taguchi method, which is known as a reliable method for improving the reliability of the result. The AFM parameters, including normal load, scratch distance, probe speed, and probe direction, are optimized on the photoresist thin film by the Taguchi method. As a result, the pinholes with diameter ranged from 25.4 ± 2.2 nm to 85.1 ± 6.3 nm, and the groove array with a depth of 40.5 ± 3.7 nm and a roughness at the bottom of less than a few nanometers was formed. Then, based on the Si/TiN/ZnO/photoresist structures, we fabricated and investigated memristors with different spot sizes and TiN top contact thickness. As a result, the HRS/LRS ratio, U*_SET_*, and I*_LRS_* are well controlled for a memristor size from 27 nm to 83 nm and ranged from ~8 to ~128, from 1.4 ± 0.1 V to 1.8 ± 0.2 V, and from (1.7 ± 0.2) × 10^−10^ A to (4.2 ± 0.6) × 10^−9^ A, respectively. Furthermore, the HRS/LRS ratio and U*_SET_* are well controlled at a TiN top contact thickness from 8.3 ± 1.1 nm to 32.4 ± 4.2 nm and ranged from ~22 to ~188 and from 1.15 ± 0.05 V to 1.62 ± 0.06 V, respectively. The results can be used in the engineering and manufacturing of memristive structures for neuromorphic applications of brain-inspired artificial intelligence systems.

## 1. Introduction

Today, the Von Neumann architecture is the basis for computing systems whose main principle is the physical separation of arithmetic logic and memory units [1,2,3,4]. Here, each stage of information processing requires several steps where data are transferred between the processor and memory [5,6]. The processing steps are carried out sequentially, imposing initial limitations on the speed of such computing devices [7,8,9,10,11,12]. This creates time and energy problems, as the information must be transferred repeatedly between different system parts. This Von Neumann bottleneck limits the future development of computing systems [13,14,15,16,17]. Some parallelism can be introduced, but it is not enough.

A promising solution is a neuromorphic architecture miming brain neuronal circuits, in which neurons represent computational units and synapses are local storage devices connected by communication channels [18,19,20,21,22]. In such an architecture, data processing is distributed throughout the computing system rather than concentrated in a central processor [23,24,25,26]. Processor and memory are integrated into a single unit, and the processing steps are performed in parallel rather than sequentially [27,28]. Although semiconductor-based computing systems have certain advantages, such as fault tolerance, power consumption, and handling of large data sets, they are significantly inferior to the biological brain [29,30,31,32]. Thus, there is a need for technology that has the advantages of biological and semiconductor materials but does not have drawbacks [33,34,35,36]. The biological brain supports various intellectual functions such as memory, learning, and decision-making [37,38,39,40]. One of the main ways of technical implementation of the biological brain is to manufacture ICs based on memristor structures, which are memory elements in the form of transition metal oxide film cells (neurons) that change their electrical resistance (between low-resistance (LRS) and high-resistance (HRS) states) under the action of an external electric field, connected by cross-synapses of data [41,42,43,44,45,46]. In doing so, ReRAM has a small cell size of a few nanometers, high integration density, high performance, and low power consumption, allowing it to mimic massive parallelism and low-power computing previously seen in the human brain [47,48,49,50,51,52]. Thus, the ReRAM technology satisfies all the basic requirements of neuromorphic systems [53,54,55].

An analysis [56,57,58,59,60,61,62] has shown that structures based on binary metal oxides are promising, especially zinc oxide (ZnO) obtained by pulsed laser deposition (PLD). To produce neuromorphic systems based on ZnO films on an industrial scale, fabrication regimes, as well as various control parameters on the resistive switching of ReRAM elements, are needed [63,64]. Therefore, it is necessary to study the effect of the ReRAM size element (the oxide film simulates the computing part of the biological brain, the neuron) and the contact thickness (the metal/oxide transition simulates the memory part of the biological brain, the synapse) on the memristive effect. This raises the need to develop new approaches for such studies, allowing local rapid prototyping of individual ReRAM elements and in situ diagnostics of their electrical and morphological parameters. A promising technique to form nanoscale structures is scratching probe nanolithography using an atomic force microscope (AFM) [65,66,67,68]. This technique involves the modification of thin polymer films by forming profiled nanoscale windows using the tip of an AFM probe. One of the ways to improve the quality of the windows is the Taguchi method, which is a statistical method developed by Genichi Taguchi, and more recently also applied to engineering, biotechnology, marketing, and advertising [69,70]. The essence of the method is to evaluate the quality performance of products and to determine whether quality losses are within the tolerance limits, as they increase as the current values of a parameter deviate from the nominal value [71]. Based on the Taguchi methods, the difference between the ideal and real objects is calculated, and the aim is to reduce it to a minimum, thereby providing an improvement in quality. So, a top contact can be formed through the windows to study the effect of different geometrical parameters on the resistive switching effect.

In this paper, we propose two different approaches to studying resistive switching in thin zinc oxide films. For this purpose, we investigated scratching probe nanolithography regimes using the Taguchi method, which is known as a reliable method for improving the reliability of the result. Based on the obtained results, prototypes of memristor structures were fabricated, and the effect of memristor size and top contact film thickness on the resistive switching was investigated.

## 2. Materials and Methods

To investigate the scratching probe nanolithography regimes, 1 × 1 cm^2^ silicon substrates were cleaned in acetone and isopropyl alcohol at temperatures of 56 °C and 83 °C, respectively, for 10 min. Then, by pulsed laser deposition (PLD) technique using a Pioneer 180 PLD module (Neocera LCC, Beltsville, MD, USA) with a KrF excimer laser (λ = 248 nm) TiN films 68.4 ± 4.2 nm thick were deposited (substrate temperature: 500 °C, target-substrate distance: 40 mm, nitrogen: 5 × 10^−3^ Torr, pulse energy: 250 mJ, laser pulse repetition rate: 10 Hz, and number of laser pulses: 30,000). Next, by PLD, ZnO films 28.3 ± 3.5 nm thick were deposited on the TiN film using a template using the following settings: substrate temperature: 300 °C, target-substrate distance: 70 mm, oxygen: 1 × 10^−3^ Torr, pulse energy: 200 mJ, laser-pulse repetition rate: 10 Hz, and number of laser pulses: 22,000.

Then, positive photoresist FP-383 and RPF-383 diluents in a 1:10 ratio were applied to the Si/TiN/ZnO structure using an ASC-100 spin coater (Yotec Instruments Co., Dongda Rd., Taiwan) at 5500 rpm. The samples were dried at 90 °C for 30 min. Morphological and electrical measurements were performed on the Ntegra scanning probe nanolaboratory (NT-MDT, Zelenograd, Russia). AFM images were obtained in the semicontact mode of AFM and processed using the software package Image Analysis 2.0 (NT-MDT, Zelenograd, Russia). The thickness of the photoresist film was found to be 40.8 ± 2.2 nm, determined by measuring a step made using classical photolithography. Scratching probe nanolithography was performed in the contact mode of AFM.

To study the indentation modes, pinholes were formed in the photoresist at normal load range from 2 μN to 8 μN and load time range from 0.1 s to 1.5 s. As a result, the normal load and load time relationships for depth and diameter were plotted. The indented pinholes were then examined by conductive AFM (CAFM) to investigate the size of the uncovered ZnO film at the bottom of the photoresist. For this purpose, the TiN film served as a ground.

The Taguchi method was used to investigate the optimal regimes of scratching probe nanolithography. The general task was to determine the AFM parameters at which the depth of the grooves is maximal and the surface roughness (*Ra*) of the bottom of the grooves (windows) is minimal. For this purpose, groove arrays (parallel to each other) were formed at different AFM parameters: normal load (8 μN, 5 μN, and 2 μN), scratch distance (50 nm, 100 nm, and 150 nm), probe speed (3 μm/s, 6 μm/s, and 9 μm/s), and probe direction (relative to the long side of the cantilever bulk: parallel and perpendicular). According to the Taguchi method, the minimum required number of experiments corresponding to four different AFM parameters with three values is 9. The orthogonal array was used for 9 experiments with different values of the AFM parameters. As a result, the signal-to-noise (S/N) ratio relationships for depth and *Ra* were plotted for each AFM parameter. To assess the statistical significance of the AFM parameters and *Ra* in depth, an analysis of variance (ANOVA) was performed using STATISTICA v.12.6 software (StatSoft, Tulsa, OK, USA) (Appendix A).

To study the effect of memristor size (*D*) on resistive switching, prototype 1 structures were made (Figure 1). Based on the obtained indentation modes, four pinholes with controlled diameters ranging from 27 to 83 nm were formed on the same sample. Then, 50 μm diameter TiN top contacts were deposited on the photoresist pinholes with open ZnO areas of different sizes. As a result, Si/TiN/ZnO/TiN structures with a memristor size (pinhole diameter) ranging from 27 to 83 nm were obtained.

To study the thickness of the effect of the top contact (H) on resistive switching, prototype 2 structures were constructed (Figure 1). For that, based on the obtained scratching probe nanolithography modes, windows without residues on the bottom were formed on four different samples with the same photoresist thickness. Then, TiN films of different thicknesses were deposited in the Si/TiN/ZnO/photoresist structures after forming the windows, and then the TiN top contacts 400 × 400 nm^2^ were formed using the lift-off process. As a result, Si/TiN/ZnO/TiN structures were obtained with a maximum contact thickness ranging from 8.3 ± 1.1 nm to 32.4 ± 4.2 nm.

Electrical measurements were made in the AFM mode. The TiN bottom film was grounded during measurements. Current–voltage curves (CVC) were obtained at a sweep voltage of −3 V to +3 V for 1000 cycles. Cumulative probabilities, U*_SET_* and I*_LRS_* vs. *H*, HRS/LRS and U*_SET_* vs. *H*, and retention tests were plotted.

## 3. Results and Discussion

Figure 2a shows the 3D AFM image of the photoresist surface that contains one of the indented pinholes. All formed pinholes were well defined. To make sure that we are not deforming ZnO film, we experimentally determined a critical normal load (at which the deformation of the ZnO film is observed) to be 9 µN. Consequently, we worked at a normal load value of up to 8 µN. Depth and diameter were well controlled at normal load from 2 to 8 µN and ranged from 12.3 ± 1.2 nm to 41.6 ± 2.5 nm in depth and 53.4 ± 4.7 nm to 86.2 ± 5.8 nm in diameter (Figure 2b). As expected, the load time ranging from 0.1 to 1.5 s had a lesser effect than the normal load and ranged from 22.1 ± 1.4 nm to 34.6 ± 2.5 nm in depth and 74.1 ± 5.5 nm to 91.3 ± 7.2 nm in diameter (Figure 2c). The non-linearity of the obtained dependencies can be explained by the viscoelastic properties of the photoresist. When the normal load was increased to 7 µN (load time at 0.5 s), the CAFM signal was silent, but at a force of 7 µN, peaks were recorded, indicating contact of the probe with the underlying ZnO film (Figure 2d). At a normal load of 8 μN, the signal broadened, which refers to an increase in the probe/ZnO size spot (*D*).

At different load times ranging from 0.1 to 1.5 s (Figure 2d inset), spots were well controlled and ranged from 25.4 ± 2.2 nm to 85.1 ± 6.3 nm (normal load of 8 μN) due to the viscoelastic properties of the photoresist when it is pierced through the thickness of the photoresist film.

Figure 3a shows a 3D AFM image of a photoresist surface that contains a groove. All formed grooves were well defined. Figure 3b shows the relationship between the width and depth of the grooves and the normal load. The depth and width were well controlled at different normal loads varying from 1 µN to 8 µN and ranging from 7.8 ± 0.8 nm to 40.5 ± 3.7 nm in depth and 47.3 ± 4.1 nm to 90.1 ± 7.6 nm in width. Figure 3c shows the relationship between the width and depth of the grooves and the probe speed. Depth and width were also well controlled at probe speeds in the range from 2 µm/s to 9 µm/s and ranged from 38.3 ± 2.4 nm to 24.1 ± 2.0 nm in depth and 91.7 ± 8.2 nm to 68.2 ± 4.7 nm in width. The decrease in width and depth with increasing probe speed can be due to the viscoelastic properties of the photoresist, as well as the fact that the probe–sample contact is weakened at high speeds, which leads to an elevation of the probe. The speed of the probe affects depth and width, albeit to a lesser extent than the normal load. Figure 3d shows the 3D AFM image of the photoresist surface carrying a window made by forming grooves parallel to each other at a certain distance. As can be seen, there is a residue of photoresist on the bottom of the window (surface roughness *Ra*). It is important to exclude or at least minimize its presence, because we plan to deposit this window with metal, and it will have a negative effect on the result. This will lead to a scattering of resistive switching parameters from device to device.

To achieve this purpose, the Taguchi method is used to optimize AFM parameters, including normal load, scratch distance, probe speed, and probe direction (Figure 4a). Here, the general task was to find the AFM parameters at which the photoresist was pierced to its full depth with a minimal surface roughness *Ra*. Therefore, the desirable results for depth and *Ra* are the largest and smallest possible values, respectively. The Taguchi method allows us to greatly reduce the number of experimental tests [68]. For our case, the orthogonal array is used; thus, only nine tests are needed. It follows from Figure 3b that a normal load of 8 µN is required to pierce the photoresist film to the full depth; however, in the Taguchi method, we set 2 µN as the minimum value, because other AFM parameters also contribute to the depth. The levels of parameters are summarized in Figure 4b, which also contains experimental results for depth and *Ra*. The depth is in the range from 7.3 ± 0.8 nm to 40.3 ± 1.5 nm, and *Ra* is in the range from 2.2 ± 1.4 nm to 14.3 ± 6.2 nm. As can be seen, the *Ra* results have a large confidence interval, which can explain the strong randomness of the photoresist ejection. Based on the formulas presented in Figure 4c,d, the values of signal-to-noise (S/N) ratios for depth and *Ra* were determined (Figure 4e,f). According to the Taguchi method, the required values of the AFM parameters correspond to the maximum S/N ratio. Therefore, we must find the values with the maximum S/N ratio within each parameter A-D, while the values of other parameters with lesser S/N ratios are not considered. Therefore, the maximum depth is achieved with the normal load of 8 μN (A1), the scratch distance of 50 nm (B1), the probe speed of 9 µm/s (C3), and the direction of the probe being perpendicular (D1(D2)). Similarly, the minimum surface roughness *Ra* is achieved with the normal load of 2 μN (A3), scratch distance of 100 nm (B2), probe speed of 6 µm/s (C2), and probe direction being perpendicular (D1(D2)). Referring to our general task of finding the AFM parameters at which the photoresist was pierced to its full depth and with minimal *Ra*, we fixed the AFM parameters at which the minimum *Ra* is observed, except the normal load, and increased it. The general task was achieved at the normal load of 5 μN (A2), scratch distance of 100 nm (B2), probe speed of 6 µm/s (C2), and probe direction being perpendicular (D1(D2)). As a result, we formed windows in the photoresist to the full film depth (40.7 ± 1.6 nm) and a roughness at the bottom of less than a few nanometers. Appendix A give the results of the analysis of variance (ANOVA) of the depth and *Ra*, respectively.

Figure 5 shows the effect of memristor size on resistive switching in ZnO films. A feature of this approach (prototype 1 in Materials and Methods) is that we can localize resistive switching quite precisely, unlike other approaches, where there is a breakdown problem across the entire top contact due to its massive dimensions [41,42,50,51,59]. Therefore, we can land an AFM probe in any area of the top contact (Figure 5a) and ensure that resistive switching occurs at a single location. Prototypes with spot sizes ranging from 27 to 83 nm showed stable bipolar resistive switching at 3V sweep voltage (Figure 5b). The cumulative probability shows quite robust HRS and LRS resistances for all prototypes (Figure 5c). The HRS/LRS ratio varied from ~8 to ~128. It is curious that the R_HRS_ is in a rather narrow range from (4.2 ± 0.3) × 10^10^ Ω to (7.3 ± 0.5) × 10^10^ Ω, whereas the R_LRS_ is in a wide range from (0.41 ± 0.05) × 10^8^ Ω to (7.2 ± 0.4) × 10^9^ Ω. This can be explained by the fact that the memristor size without oxygen vacancies (HRS) does not strongly affect the resistance, based on previous studies of the resistive switching mechanism in zinc oxide films [57,58,59], according to which nanoscale conduction channels of oxygen vacancies are located along the grain boundary (filamentary mechanism); however, after resistive switching (LRS), the memristor size is proportional to the number of oxygen vacancies, and, accordingly, in this case, the resistance deviation increases. The SET voltage (U*_SET_*) and the LRS current (I*_LRS_*) are well controlled for the memristor size from 27 nm to 83 nm and ranged from 1.4 ± 0.1 V to 1.8 ± 0.2 V and (1.7 ± 0.2) × 10^−10^ A to (4.2 ± 0.6) × 10^−9^ A, respectively (Figure 5d). LRS decrease with the increase in memristor size can be linked to the filament increase in width (like a dendrite) with an increase in the memristor size, and hence the LRS decreases [57,59]. It is also worth considering that with an increase in the memristor size, the probability of the occurrence of several filaments increases [55]. The retention test showed that R_HRS_ and R_LRS_ persist for at least 10,000 s for all prototypes. (Figure 5e).

Figure 6 shows the effect of top contact thickness (*H*) on the resistive switching in ZnO films. In this approach (prototype 2 in Materials and Methods), we cannot claim exactly where the resistive switching takes place, but we can quite precisely determine the thickness of the top contact using the AFM method (Figure 6a,b). The ripped edges of the TiN structures are the result of the lift-off process and do not affect the study of the impact of *H* on resistive switching. Prototypes with *H* ranging from 8.3 ± 1.1 nm to 32.4 ± 4.2 nm showed a stable bipolar resistive switching at 3V sweep voltage (Figure 6c). The HRS/LRS ratio varied from ~24 to ~186. The cumulative probability was quite robust, with the best result in the prototype with a TiN film thickness of *H* = 32.4 ± 4.2 nm. R_LRS_, R_HRS_, and R_HRS_/R_LRS_, in this case, are (3.1 ± 0.3) × 10^8^ Ω, (1.1 ± 0.4) × 10^11^ Ω, and 192, respectively (Figure 6d). The U*_SET_* and HRS/LRS ratios are well controlled at TiN top contact thickness from 8.3 ± 1.1 nm to 32.4 ± 4.2 nm and ranged from 1.15 ± 0.05 V to 1.62 ± 0.06 V and ~22 to ~188, respectively (Figure 6d). This can be explained by the fact that TiN is a reservoir of oxygen vacancies [72,73]. Therefore, an increase in the TiN top-contact thickness leads to an increase in the number of oxygen vacancies involved in resistive switching. This leads to an increase in the conductive paths and, consequently, a decrease in the I*_LRS_* (Figure 6c). However, when the oxygen vacancies are concentrated near the ZnO/TiN interface as well as in the TiN volume (HRS), the resistance of the ZnO film should not differ significantly at different *H* values, as confirmed by the experimental results. It is also worth noting that a higher number of vacancies requires a higher switching voltage. The retention test showed that R_HRS_ and R_LRS_ persist for at least 10,000 s for all prototypes. For *H* = 32.4 ± 4.2 nm, the HRS/LRS ratio varied from 192 to 181 (Figure 6e).

Thus, increasing the size of the memristor (*D*) and the top contact thickness (*H*) leads to an increase in the HRS/LRS ratio, an increase in the U_SET_, and an increase in the I*_LRS_.* Unfortunately, a ‘universal’ memory does not exist, so we should proceed from the issue: when designing a neuromorphic system, in one case, multilevel characteristics may be a priority, and in another case, power consumption may be a priority. Hence, they should be based on the task: if the HRS/LRS ratio is a priority (for multilevel), *D* and *H* should be increased; if power consumption is a priority, they should be decreased.

The results obtained from this study of resistive switching parameters correlate well with the results presented in [74], which present ZnO-based RRAM devices fabricated by pulsed laser deposition methods.

## 4. Conclusions

In conclusion, we optimized the scratching probe nanolithography of the photoresist for the intended pinholes and grooves array. For intended pinholes, the depth and diameter are well controlled at different average loads varied from 2 µN to 8 µN and ranged from 12.3 ± 1.2 nm to 41.6 ± 2.5 nm in depth and 53.4 ± 4.7 nm to 86.2 ± 5.8 nm in diameter. Furthermore, depth and diameter are well controlled at load times that ranged from 0.1 to 1.5 s and ranged from 22.1 ± 1.4 nm to 34.6 ± 2.5 nm in depth and 74.1 ± 5.5 nm to 91.3 ± 7.2 nm in diameter. The Taguchi method is employed to optimize AFM parameters (including normal load, scratch distance, probe speed, and probe direction) for the grooves array. As a result, we formed windows in the photoresist to full film depth (40.7 ± 1.6 nm) and roughness at the bottom of less than 1 nm.

The results of scratching probe nanolithography were used for two memristive structure approaches. The first approach (prototype 1) was used to study the size of the memristor on resistive switching. Prototypes with memristor sizes ranging from 27 nm to 83 nm showed stable bipolar resistive switching with HRS/LRS ratios ranging from ~8 to ~128. Additionally, U*_SET_* and I*_LRS_* are well controlled and ranged from 1.4 ± 0.1 V to 1.8 ± 0.2 V and (1.7 ± 0.2) × 10^−10^ A to (4.2 ± 0.6) × 10^−9^ A, respectively. The retention test showed that R_HRS_ and R_LRS_ persist for at least 10,000 s. The second approach (prototype 2) was used to study the top contact thickness on the resistive switching. Prototypes with *H* ranging from 8.3 ± 1.1 nm to 32.4 ± 4.2 nm showed a stable bipolar resistive switching at 3V with an HRS/LRS ratio varying from ~24 to ~186. In addition, U*_SET_* was well controlled and ranged from 1.15 ± 0.05 V to 1.62 ± 0.06 V. The retention test also showed that R_HRS_ and R_LRS_ persisted for at least 10,000 s. The results can be used in engineering and manufacturing memristive structures for neuromorphic applications of brain-inspired artificial intelligence systems.

## Figures and Tables

**Figure 1 nanomaterials-13-01583-f001:**
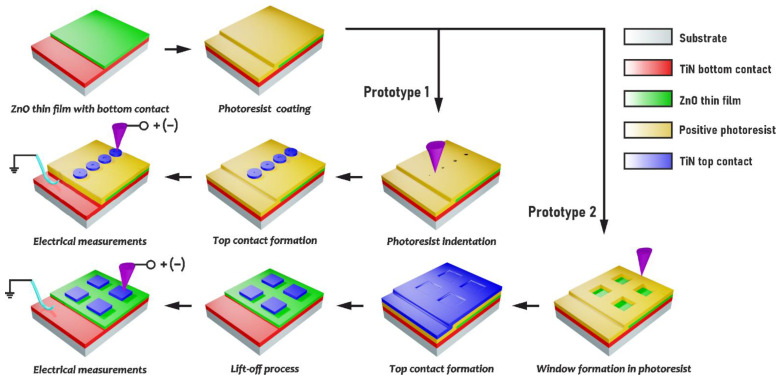
Process flow diagram of the memristor structures to investigate the effect of memristor size (prototype 1) and top contact thickness (prototype 2) on resistive switching.

**Figure 2 nanomaterials-13-01583-f002:**
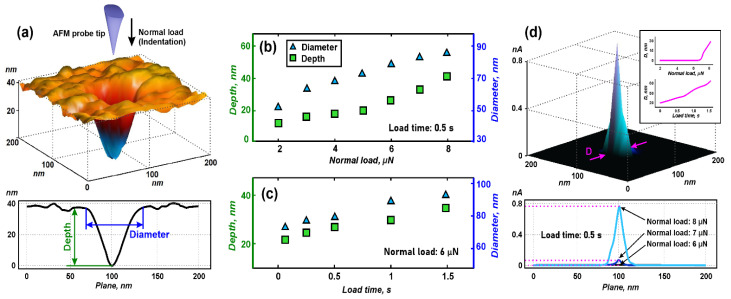
Study of scratching probe nanolithography on photoresist: (**a**)—3D-AFM image and AFM cross-section of the single indented pinhole; (**b**)—dependences of depth and diameter on normal load; (**c**)—dependences of depth and diameter on and load time; (**d**)—CAFM-image of the single indented pinhole and AFM cross-sections at different normal loads.

**Figure 3 nanomaterials-13-01583-f003:**
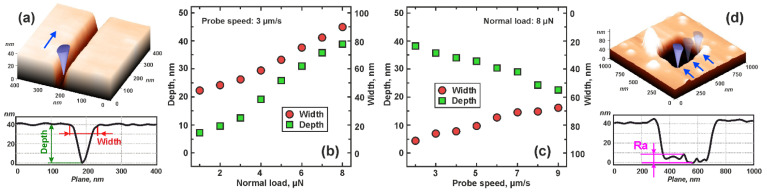
Study of scratching probe nanolithography on photoresist: (**a**)—3D-AFM image and AFM cross section of the groove; (**b**)—dependences of the photoresist puncture depth and structure width on the normal load; (**c**)—dependences of the photoresist puncture depth and structure width on the scan speed; (**d**)—3D-AFM image and AFM cross section of a window.

**Figure 4 nanomaterials-13-01583-f004:**
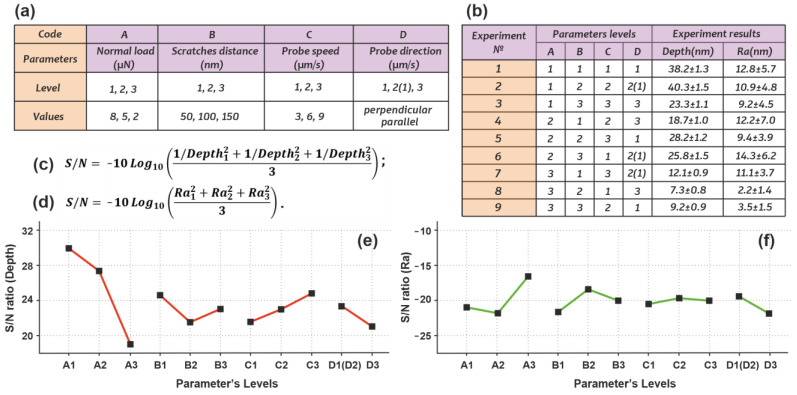
Study of depth and surface roughness using the Taguchi method: (**a**)—AFM parameters and their levels; (**b**)—L9 orthogonal array; (**c**,**e**)—S/N ratios for depth for each parameter; (**d**,**f**)—S/N ratios for *Ra* for each parameter.

**Figure 5 nanomaterials-13-01583-f005:**
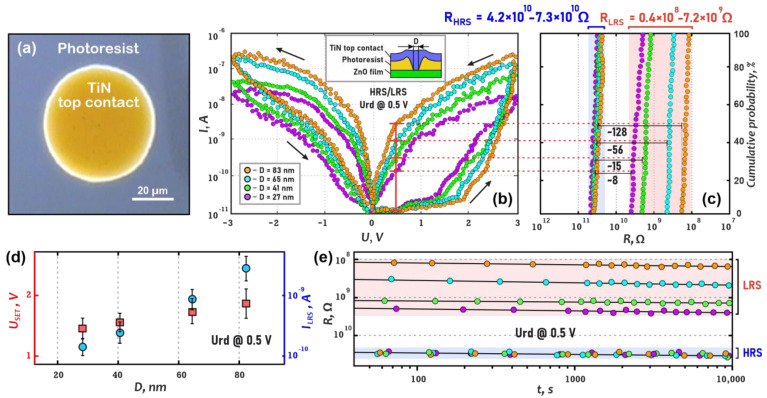
Study of the memristor size (D) on resistive switching in the Si/TiN/ZnO/TiN structure: (**a**)—microphotography; (**b**)—current-voltage characteristics; (**c**)—cumulative probability; (**d**)—U_SET_ and I*_LRS_* vs. D; (**e**)—retention test.

**Figure 6 nanomaterials-13-01583-f006:**
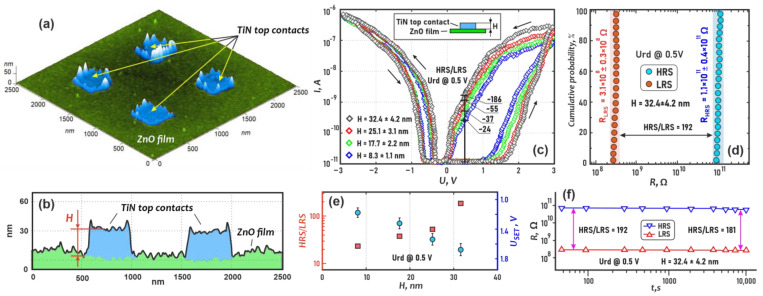
Study of the top contact thickness (*H*) on resistive switching in the Si/TiN/ZnO/TiN structure: (**a**)—3D-AFM image of ZnO film and TiN electrodes; (**b**)—AFM cross-section; (**c**)–current-voltage characteristics; (**d**)—cumulative probability; (**e**)—HRS/LRS ratio and U_SET_ vs. *H*; (**f**)—retention test.

## Data Availability

Not applicable.

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
