# Peer review of "Approaches for Memristive Structures Using Scratching Probe Nanolithography: Towards Neuromorphic Applications"

_nanomaterials, 2023, doi:10.3390/nano13101583_

Round 1
Reviewer 1 Report
The manuscript investigated the optimization of the scratching probe nanolithography on the photoresist for intended pinholes and grooves array. The scratching probe nanolithography were also developed for constructing two memristive device structures to study the size and electrode thickness on the resistive switching characteristics. The work proposed a method for developing memristive device. Some issues should be addressed before possible publication.
1. The switching mechanism of the memristive device was not mentioned in the manuscript. It is suggested to give a brief description on this toptic.
2. It is suggestted to make a comparsion on the switching performance of present device and other ZnO-based memristive devices?
3. It is well known that conductive filament size is independent on the device size,why the LRS decrease with the increase of memristor size?
Reviewer 2 Report
In this paper authors presented interesting and good results on memristive structures using scratching probe nanolithography and highlighting its potential application in neuromorphic systems.
The reviewer suggests the authors to make minor revision to this version of the manuscript considering the below remarks/comments and suggestions. These changes do not substantially affect the quality of the manuscript, but help to improve the quality and presentation of the work. The suggested changes can be incorporated in the revised version of the manuscript.
Comments:
· As Taguchi method is used in optimization process of AFM parameters, it is good to explain this part in detail by quoting relevant references.
· In line no. 87, 90, 93, the temperature units are not considered properly. Be consistent with the units throughout.
· In Fig. 2(c), it can be seen that the width increases with probe speed. Explain the reason for the observation this trend.
· The explanation for the results shown in Figs. 4(e) and (f) are not very clear for the different S/N ratio trend observed. Provide more details. Some parameter details missing in the text. For ex: B3.
· The minimum size of the memristor considered is 27nm (diameter). Indicate the term ‘diameter’ in the text while mentioning the size of the device. Any experimental limitations to go less than 27nm? How is the switching behaviour of the device if the size is less than 27nm?
· For mentioning SET voltage, it is better to use consistent notation as in widely reported articles.
· Use appropriate abbreviations.
· Check for typos and grammatical errors
Reviewer 3 Report
Comments:
1. Some relevant papers in ZnO-based resistive switching field should be mentioned and cited in the section of introduction. Therefore, I strongly recommend the authors to revise this part. Such as some relevant references listed as follows: 1. Nanomaterials 12 (2022): 455, doi.org/10.3390/nano12030455; 2. Ceramics International, 43 (2017): S474-S480, doi.org/10.1016/j.ceramint.2017.05.213; 3. RSC advances, 7 (2017): 38757-38764, doi.org/10.1039/C7RA07100K.
2. Microstructure test and analysis were missed. Thus, XRD or XPS test should be performed to better characterize of properties in devices.
3. What about the switching speeds characteristics in devices? This characteristic in RS devices is crucial to evaluate RS performance. I strongly recommend the authors need to add this characteristic.
4. The authors should make a comparison with the results in recent publications.
5. The authors should explain the resistive switching mechanism.
6. Finally, there are some grammar mistakes and confusing sentences in the manuscript. The English level of the manuscript need to improve. Please ask a native English speaker to revise and proofread their revised manuscript before re-submission.
There are some grammar mistakes and confusing sentences in the manuscript. The English level of the manuscript need to improve. Please ask a native English speaker to revise and proofread their revised manuscript before re-submission.
Reviewer 4 Report
See attachment for my feedback

Round 2
Reviewer 1 Report
The authors have addressed the concerns.